# Effect of Korean Medicine Treatment on Patients with Postherpetic Neuralgia: A Retrospective Chart Review

**DOI:** 10.3390/healthcare12020256

**Published:** 2024-01-19

**Authors:** Hyoseung Jeon, Suji Lee, Sung-A Kim, Unhyung Lee, Seunghoon Lee

**Affiliations:** 1Department of Acupuncture and Moxibustion Medicine, Kyung Hee University Medical Center, Seoul 02447, Republic of Koreasjstarry41@naver.com (S.L.);; 2Department of Clinical Korean Medicine, Graduate School, Kyung Hee University, Seoul 02453, Republic of Korea; 3Department of Acupuncture and Moxibustion, College of Korean Medicine, Kyung Hee University, Seoul 02453, Republic of Korea

**Keywords:** Korean medicine treatment, pain intensity, postherpetic neuralgia, quality of life, retrospective chart review

## Abstract

Evidence regarding Korean medicine treatment (KMT) for neuropathic pain is lacking. We aimed to identify the effects of integrative KMT in patients with postherpetic neuralgia (PHN). We retrospectively analyzed the electronic medical records of patients with PHN who received KMT at Kyung Hee University Korean Medicine Hospital between August 2021 and July 2022. We evaluated the effects of KMT—comprising acupuncture, pharmacopuncture, herbal medicine, cupping, and moxibustion—on pain intensity using the numerical rating scale (NRS), Short-Form McGill Pain Questionnaire (SF-MPQ), Hospital Anxiety and Depression Scale–Anxiety (HADS-A), Hospital Anxiety and Depression Scale–Depression (HADS-D), Daily Sleep Interference Scale (DSIS), Fatigue Severity Scale (FSS), and EuroQol-5D. Among 53 patients with PHN, 13 were included. The NRS score for worst pain over 1 week decreased from 6.54 ± 0.64 at baseline to 3.85 ± 0.63 at 8 weeks (41% reduction, *p* < 0.01), while that for average pain over 1 week decreased from 4.93 ± 0.67 at baseline to 3.08 ± 0.46 at 8 weeks (37% reduction, *p* < 0.01). From baseline to 8 weeks, there were significant reductions in the SF-MPQ, HADS-A, FSS, and EuroQol-5D scores. No adverse events were reported after KMT. Therefore, KMT may be an effective treatment option for patients with PHN.

## 1. Introduction

Herpes zoster (HZ) is an acute disease caused by the varicella zoster virus (VZV) and is accompanied by a painful rash. Postherpetic neuralgia (PHN) is defined as pain persisting for ≥3 months after the resolution of an acute HZ rash [1]. As a complication of HZ, PHN is characterized by neuropathic pain such as throbbing, aching, burning, shooting pain, hyperalgesia, allodynia, and paresthesia in the mucocutaneous area, whereas VZV causes eruptions.

The lifetime prevalence of HZ is 11.5% [2], and it is estimated that 5–20% of those with HZ develop PHN [3]. In a large epidemiological study [4] conducted from 1994 to 2018, the overall incidence of HZ and PHN increased continuously every year in the United States.

For patients with PHN, as the pain becomes chronic, it is likely that the patients’ quality of life (QoL) decreases due to pain and comorbidities, including sleep disturbance and emotional problems such as anxiety and depression [5]. PHN, which causes both physical and mental problems, can have serious socioeconomic consequences.

Currently, the first-line therapy for PHN is the administration of medications including serotonin and noradrenaline reuptake inhibitors, tricyclic antidepressants, opioids, antiepileptic drugs, and topical medications [6]. However, many patients do not respond to these medications, and several adverse reactions, such as nausea, fatigue, drowsiness, and dizziness, have been reported [7]. In particular, elderly patients most frequently affected by PHN are at risk of falls and hip fractures due to orthostatic hypotension when taking tricyclic antidepressants [8].

Acupuncture treatment, pharmacopuncture treatment, herbal medicine (HM), cupping treatment, and moxibustion treatment are widely used as alternatives to conventional treatments for pain diseases. There is considerable clinical evidence that acupuncture [9], pharmacopuncture [10,11], and HM [12,13] are effective in improving pain. Several studies have proposed the mechanisms of acupuncture for analgesic effect. The local analgesic effects of acupuncture may be mediated by adenosine A1 receptors [14] or myofascial trigger point inactivation [15]. Acupuncture can also induce a segmentally analgesic effect via the gate control theory of pain. Acupuncture stimulation on some traditional acupuncture points located at the extremities can generally reduce pain through the activation of descending inhibition by releasing serotonin and noradrenaline or diffuse noxious inhibitory control [16]. One systematic review [17] noted that acupuncture can relieve pain in trigeminal neuralgia. Bee venom acupuncture (BVA) is often used for neuropathic pain because of its analgesic, anti-inflammatory, and nerve-protection effects [18,19,20]. Some case reports [21,22,23] showed that BVA has the potential to be an effective treatment for patients with neuralgia. Some HMs have neuroprotective effects [24], inhibit spinal pain transmission [25], decrease the production of inflammatory cytokines, and reduce inflammatory reactions [26]. Oral HMs are frequently used to treat PHN in practice [27]. It has also been reported that moxibustion has a blood circulation-promoting effect [28] and an analgesic effect [29]. Bleeding cupping therapy can reduce local serum P substance content in patients with PHN, resulting in an analgesic effect [30]. One randomized controlled trial [31] showed that moxibustion plus cupping therapy was effective in treating trigeminal neuralgia.

Since chronic pain diseases are accompanied by various symptoms as well as the pain itself, in Korea, these treatments are mainly used clinically in the form of integrative Korean medicine treatment (KMT) by combining these treatments. Integrative KMT, especially, is frequently used in clinical practice to treat the complex symptoms of PHN [32]. Many clinical studies are confined to nociceptive pain, but there is still a lack of evidence regarding neuropathic pain [33,34]. Although there are some case studies and experimental studies [35,36], more clinical research is needed on the effectiveness and safety of KMT for PHN. Therefore, in this study, we aimed to observe the effectiveness and safety of integrative KMT on PHN by retrospectively reviewing the records of patients with PHN regarding pain intensity, anxiety, depression, sleep disturbance, fatigue, quality of life, and adverse events.

## 2. Materials and Methods

### 2.1. Ethics Statement

This study was approved by the Institutional Review Board of Kyung Hee University Korean Medicine Hospital (KOMCIRB 2022-07-003). Given the retrospective nature of this study and the use of anonymized patient data, the requirement for informed consent was waived.

### 2.2. Study Design and Inclusion Criteria

The electronic medical records (EMRs) of patients with PHN who received outpatient or inpatient treatment at the Department of Acupuncture and Moxibustion, Kyung Hee University Korean Medicine Hospital (KHUKMC), between August 2021 and July 2022 were retrospectively reviewed. Potential participants were identified by searching for the International Statistical Classification of Diseases and Related Health Problems, Tenth revision (ICD-10) codes related to PHN (G53.0, G63.0, or B02.2) or HZ (B02.8 or B02.9) in the EMRs, and patients who met the following inclusion criteria were finally selected: an age ≥19 years; the presence of PHN with pain lasting >3 months after the resolution of a rash and with the worst pain rated as ≥4 on the numerical rating scale (NRS); and treatments performed more than twice, pain or other symptom scales used to evaluate PHN specified, and the results recorded in the EMR. Patients whose pain was related to diseases other than PHN or who visited other hospitals and underwent additional pharmacotherapy for PHN as well as those with other chronic diseases that may interfere with KMT effects were excluded.

### 2.3. Integrative Korean Medicine Treatments

Patients with PHN were provided semi-standardized KMT twice a week at the Department of Acupuncture and Moxibustion, Kyung Hee University Korean Medicine Hospital (Figure 1). Detailed treatments were as follows.

#### 2.3.1. Acupuncture Treatment

Acupuncture treatment was administered using 0.25 × 40 mm disposable sterilized needles (Dongbang Acupuncture, Inc., Chungnam, Republic of Korea). The selection of 1 to 8 acupuncture points was based on traditional acupuncture theory and neurophysiological perspectives [37], categorizing points into local, segmental, or distal acupuncture points. Local points, which include classical acupuncture, tender, or trigger points, are situated near the painful region. Segmental points involve selecting points innervated by the same meridian or spinal nerve as the painful area, exemplified by EX-B2 on paraspinal muscles. Distal points, chosen on the contralateral side of the body or extremities, address general symptoms accompanying chronic pain, such as insomnia (PC6, HT7, BL62, or KI6), depression/anxiety (PC6, HT7, LI4, or SP6), or fatigue (ST36, SP6, LU8, or GB20). The needling was maintained for 15 min during the acupuncture session.

Electrical stimulation was added at up to four points among the previously mentioned acupuncture points. Using an electrical acupuncture device (STN-330; StraTek Co., Ltd., Anyang, Republic of Korea), up to 80% of the maximum intensity tolerated by the patient was applied at 2 Hz at local points, 100 Hz at segmental points, and 2/4 Hz at distal points (2 changes every second).

#### 2.3.2. Bee Venom Acupuncture Treatment

Following a negative result from an allergic reaction skin test, 0.5 mL of bee venom at a concentration of 1:30,000 was administered to the same local or segmental point as the acupuncture treatment using a 1 mL disposable syringe (30 gauge, Hwajin Medical Co., Seoul, Republic of Korea). To achieve this concentration, 10 mg of dried BV (Yoomil Garden, Hwasun, Republic of Korea) was diluted with 300 mL of saline (Joongwe Pharmaceuticals, Seoul, Republic of Korea).

#### 2.3.3. Herbal Medicine

The Korean medicine doctor prescribed Sosiho-tang, Oryeongsan, and Gamiguibi-tang according to the pattern identified by considering the patient’s accompanying symptoms. Soshiho-tang was prescribed to alleviate hyperactivity in the sympathetic nervous system by reducing liver Qi stagnation, Oryeongsan was used to address body fluid homeostasis imbalance, including edema and urinary dysfunction, and Gamiguibi-tang was administered for symptoms of anxiety and insomnia related to heart and spleen deficiency. All herbal medicines were manufactured by Kracie Pharma Korea Co., Ltd. (Seoul, Republic of Korea) in the form of extract fine granules, and were prescribed to be taken as one pack (3 g: Sosiho-tang and Oryeongsan; 3.75 g: Gamiguibi-tang) three times a day.

#### 2.3.4. Cupping Treatment

Cupping treatment using a Hansol cupping machine (Hansol Medical Co., Ltd., Paju, Republic of Korea) was administered on the muscles along the spine as a segmental point for 5 min after acupuncture treatment.

#### 2.3.5. Moxibustion Treatment

An electronic moxibustion device (Ontum; TechnoScience Co., Ltd., Seoul, Republic of Korea) was applied to the area that felt painful or cold for 15 min simultaneously with acupuncture treatment.

### 2.4. Data Collection

The EMRs of the eligible patients were reviewed, and the following data were collected: the patient characteristics (sex, age, height, and weight), treatment modality (acupuncture, pharmacopuncture, HM, cupping, and moxibustion), total number of treatments, frequency of KMT, anamnesis, medication history, invasive therapy history, the period from the date of HZ onset, regions of rashes, and scores for pain (NRS-A, NRS-W, and Short-Form McGill Pain Questionnaire (SF-MPQ)), anxiety and depression (Hospital Anxiety and Depression Scale (HADS)), and SF-MPQ), sleep disturbance (Daily Sleep Interference Scale (DSIS)), fatigue (Fatigue Severity Scale (FSS)), and QoL (EuroQol-5D (EQ-5D)) at each measurement time point.

### 2.5. Outcomes

The first outcome was measured before treatment, when the patient first visited the hospital. Outcomes were also measured at 2, 4, 6, and 8 weeks after KMT. The Korean medical doctor assessed the outcomes by providing a pre-designed questionnaire, and the patient filled out all of the questionnaires in front of the doctor. If patients had questions about the questionnaire, they asked the Korean medical doctor and filled it out. If the outcomes were measured between two time points, the measurements were allocated to the closet time point. Scale measurements obtained until 6 days after the first visit were excluded from the results of week 2, and those obtained 9 weeks after the first visit were excluded from the results of week 8.

#### 2.5.1. Primary Outcome

The primary outcome was a pain intensity measured using the NRS. The NRS is a subjective measure of pain level rated on a 0–10 scale, with 0 representing no pain and 10 representing the worst pain imaginable. The average pain scores over 1 week (NRS-A) and worst pain scores over 1 week (NRS-W) were recorded.

#### 2.5.2. Secondary Outcome

##### Multidimensional Qualities of Pain

The multidimensional qualities of pain for patients with PHN were measured using the SF-MPQ, which consists of 15 descriptors [38]: 11 descriptors represent the sensory dimension of pain experience (SF-MPQ-S), and 4 descriptors represent the affective dimension of pain (SF-MPQ-A). Each of the 15 descriptors was ranked by the patient on a 4-point intensity scale (0 = none, 1 = mild, 2 = moderate, and 3 = severe).

##### Anxiety and Depression

The HADS measures the symptoms of anxiety and depression [39]. It consists of 14 questions and is divided into 7 questions each in the subsections of anxiety (HADS-A) and depression (HADS-D). The scores for each subsection are summed and analyzed separately. For each subsection, a score from 0 to 7 is classified as normal, 8 to 10 as mild depression or anxiety, 11 to 14 as moderate anxiety or depression, and 15 to 21 as severe depression or anxiety.

##### Sleep Disturbance

The sleep disturbance of patients with PHN was assessed using the DSIS, which consists of an 11-point Likert scale on which patients assess their sleep condition during the past 24 h [40]. Response options range from 0 (did not interfere with sleep) to 10 (completely interfered with sleep).

##### Fatigue

The FSS is used to assess fatigue and consists of 9 questions [41]. The FSS score is the average of the item scores based on a 7-point Likert scale for each question. Lower scores indicate less fatigue, whereas higher scores indicate more fatigue.

##### Quality of Life

QoL was measured using the EQ-5D includes a health status classification system with 5 dimensions (mobility, self-care, usual activities, pain/discomfort, and anxiety/depression), and each dimension has 5 levels (1 = none, 2 = mild, 3 = moderate, 4 = severe, or 5 = extreme) [42].

##### Adverse Events

Generally, there are adverse events (AEs) related to acupuncture, including subcutaneous hematoma, skin bruising, and needle site pain [43]. AEs of pharmacopuncture include itching, swelling, and anaphylaxis [44]. HM can cause insomnia, nervousness, tremors, and headaches.

In the EMRs, we searched for AEs confirmed during the treatment period, and their relationship with the treatment was considered. In the case of HM, the number of HMs returned due to AEs was collected.

### 2.6. Subgroup Analysis

To identify the conditions under which KMT is more effective in relieving pain caused by PHN, subgroup analyses were conducted according to the region of pain and the session of treatment. First, participants were divided into a group affected on the trunk and a group affected on the face. In addition, based on the treatment administered in the first 4 weeks, participants were divided into a group who received treatment more than twice a week and a group who received treatment less than twice a week.

### 2.7. Statistical Analysis

Continuous variables at each time point are presented as mean ± standard deviation or mean ± standard error, and categorical variables are presented as numbers and percentages. The last observation carried forward method was used to handle the missing follow-up data. All outcomes except the AEs to be measured correspond to continuous variables, and normality was tested using the Shapiro–Wilk test. If it was normally distributed, it was analyzed using the paired t-test. Otherwise, it was analyzed using the Wilcoxon signed-rank test. AEs were summarized according to the incidence proportion. Statistical analyses were performed using SPSS software (version 25.0; IBM Corporation, Armonk, NY, USA). A *p*-value < 0.05 was considered significant.

## 3. Results

From the EMRs, the data of 53 patients with a record of visiting the KHUKMC Department of Acupuncture and Moxibustion and having a principal diagnosis of ICD-10 codes related to PHN or HZ between August 2021 and July 2022 were extracted. Charts of patients with ICD-10 codes related to HZ were carefully identified to determine whether pain after the resolution of the rash had been present for >3 months. After screening, 13 patients (6 men, 7 women) who satisfied the inclusion criteria were included in the final study (Figure 2).

The patients’ mean age was 70.38 ± 10.68 years, and the mean duration of pain was 27.08 ± 19.56 months. All patients received pharmacotherapy more than 3 months before receiving KMT for treating symptoms of PHN. One Korean medical doctor (S. Lee) treated all patients for over 8 weeks, but only results up to 8 weeks were observed. Patients were advised to receive KMT twice a week, but the patients received KMT approximately 12 times over 8 weeks. The detailed demographics and clinical characteristics of the study patients are summarized in Table 1.

Among the administered KMTs, acupuncture and BVA were prescribed to all patients. HM, wet cupping, and moxibustion were administered to 9 (69%), 10 (77%), and 11 (85%) patients, respectively (Table 2).

The average pain and worst pain over a week as the primary outcomes were assessed using the NRS (Figure 3). The worst pain scores were 6.54 ± 0.64 at baseline, 5.46 ± 0.81 at 2 weeks, 4.3 ± 0.64 at 4 weeks, 3.92 ± 0.60 at 6 weeks, and 3.85 ± 0.63 at 8 weeks. The reduction in worst pain was statistically significant at 4, 6, and 8 weeks compared to baseline (34% pain reduction at 4 weeks, *p* = 0.002; 40% pain reduction at 6 weeks, *p* = 0.002; 41% pain reduction at 8 weeks, *p* = 0.002). The average pain scores were 4.93 ± 0.67 at baseline, 4.07 ± 0.58 at 2 weeks, 3.38 ± 0.45 at 4 weeks, 3.23 ± 0.39 at 6 weeks, and 3.08 ± 0.46 at 8 weeks. The average pain reductions observed at 2, 4, 6, and 8 weeks compared to baseline were statistically significant (17% pain reduction at 2 weeks, *p* = 0.026; 31% pain reduction at 4 weeks, *p* = 0.004; 34% pain reduction at 6 weeks, *p* = 0.004; 37% pain reduction at 8 weeks, *p* = 0.005).

The secondary outcomes are presented in Table 3. With regard to the SF-MPQ (SF-MPQ-S and SF-MPQ-A, respectively), the scores were 33.85 ± 10.18 (26 ± 7.27 and 7.85 ± 3.24) at baseline and 26.08 ± 8.15 (20.85 ± 7.15 and 5.31 ± 1.55) at 8 weeks (23% reduction in the SF-MPQ score, *p* < 0.01; 20% reduction in the SF-MPQ-S, *p* < 0.01; 32% reduction in the SF-MPQ-A score, *p* < 0.05). Although the HADS-A scores were 8 ± 6.42 at baseline and 2.62 ± 2.22 at 8 weeks (67% reduction in the HADS-A score, *p* < 0.01), the HADS-D scores were 7.77 ± 5.36 at baseline and 5.54 ± 3.2 at 8 weeks (29% reduction in the HADS-D score, *p* = 0.115). The DSIS scores were 3.38 ± 2.72 at baseline and 2 ± 2.13 at 8 weeks (41% reduction in the DSIS score, *p* = 0.172). The FSS scores were 3.7 ± 1.63 at baseline and 2.16 ± 1.38 at 8 weeks (42% reduction in the FSS score, *p* < 0.01). The EQ-5D scores were 5.31 ± 3.1 at baseline and 3.15 ± 1.72 at 8 weeks (41% reduction in the EQ-5D score, *p* < 0.05). Compared with baseline, the SF-MPQ scores were significantly decreased at 2, 4, 6, and 8 weeks (SF-MPQ-S and SF-MPQ-A), and the HADS-A, FSS, and EQ-5D scores were significantly decreased at 4, 6, and 8 weeks. No significant changes were observed at any time point in the HADS-D and DSIS scores.

In the subgroup analysis according to the region of the rash, the group affected on the trunk showed scores of 6.75 ± 0.82 on the NRS-W and 5.13 ± 0.88 on the NRS-A at baseline and 3.13 ± 0.44 on the NRS-W and 2.88 ± 0.4 on the NRS-A at 8 weeks (54% reduction in the NRS-W score, *p* < 0.05; 44% reduction in the NRS-A score, *p* < 0.05). The group affected on the face showed scores of 6.2 ± 1.11 on the NRS-W and 4.6 ± 1.17 on the NRS-A at baseline and 5 ± 1.41 on the NRS-W and 3.4 ± 1.08 on the NRS-A at 8 weeks (19% reduction in the NRS-W score, *p* = 0.059; 26% reduction in the NRS-A score, *p* = 0.063; Figure 4).

The data of the patients were also analyzed based on the number of treatment sessions. The group that received KMT more than twice a week for the first 4 weeks tended to show greater improvement in pain symptoms than the group that received KMT less than twice a week for the first 4 weeks. In the group receiving KMT more than twice a week, scores were 6.6 ± 0.98 on the NRS-W and 4.8 ± 1.2 on the NRS-A at baseline and 3.4 ± 0.93 on the NRS-W and 3 ± 0.89 on the NRS-A at 8 weeks (48% reduction in the NRS-W score, *p* < 0.05; 38% reduction in the NRS-A score, *p* < 0.05). In the group receiving KMT less than twice a week, scores were 6.5 ± 0.89 on the NRS-W and 5 ± 0.87 on the NRS-A at baseline and 4.13 ± 0.88 on the NRS-W and 3.13 ± 0.55 on the NRS-A at 8 weeks (36% reduction in the NRS-W score, *p* < 0.05; 37% reduction in the NRS-A score, *p* < 0.05; Figure 5). Among the 13 patients, there were no returned HMs due to AEs (0%), and there were no records of any other mild or severe AEs (0%).

## 4. Discussion

This study aimed to identify the effect of KMT by retrospectively reviewing the EMRs of patients with PHN. KMT has been shown to reduce both the worst and average pain levels in patients with PHN. It showed worst pain reductions of 2.23 (34%) at 4 weeks and 2.69 ± 0.01 (41%) at 8 weeks when measured using the NRS. Additionally, this study revealed average pain reductions of 1.54 ± 0.22 (31%) at 4 weeks and 1.84 ± 0.21 (37%) at 8 weeks when measured using the NRS. These results indicate a meaningful pain reduction, considering that an approximately 2-point or 30% reduction in the NRS score indicates clinically significant pain relief in patients with chronic disease [45], and the degree of pain reduction in this study was similar to that experienced when using Western treatments such as pregabalin [46], gabapentin [47], pulsed radiofrequency [48], and lidocaine–stellate ganglion blockade [49]. In addition, similar to the results of previous studies [46,47,48,49], the pain decreased rapidly until 4 weeks, and then decreased gradually until 8 weeks.

It is notable that, in addition to pain, the present study showed an improvement in anxiety, depression, sleep disorder, fatigue, and QoL after 8 weeks of KMT. Through score reductions of 23% in the SF-MPQ, 20% in the SF-MPQ-S, and 32% in the SF-MPQ-A at 8 weeks compared to those at the baseline, it was found that KMT improved not only pain, but also emotional disorders.

Regarding emotional disorders, anxiety measured using the HADS-A tended to be more improved than depression measured using the HADS-D. Chronic pain is more directly related to anxiety than to depression by affecting the sympathetic nerve activity [50,51]. Similarly, this study showed that KMT is effective not only in improving chronic pain, but also in improving anxiety. Chronic pain and depression are also closely related [52,53]; however, according to previous studies, long-term treatment is often needed to improve depression [54,55,56]. In this study, only the short-term treatment effect could be investigated, so there was not much improvement in depression.

A previous study [57] showed that acupuncture was effective in patients with chronic pain-related insomnia. However, in this study, there was no meaningful improvement in sleep disorders, as measured using the DSIS. For the DSIS, a score <4 indicates that sleep disturbance is classified as mild [58]. The poor improvement in the DSIS score in this study could be attributed to a mild degree of sleep disorders at baseline (3.38 ± 2.72).

To identify the effect of KMT on PHN in detail, subgroup analysis was performed by dividing the groups according to the rash region and treatment sessions. Studies [59,60] on the status of PHN showed that the most common bodily region affected by rashes was the trunk, followed by the face. All patients included in our study had been affected on the trunk (8 patients, 62%) or face (5 patients, 38%). In this study, patients affected on the trunk tended to show a greater improvement in pain than those affected on the face. An experimental study [61] suggested that, as the critical node in the affective pain circuit, the lateral parabrachial nucleus is activated more strongly by noxious stimulation of the face than of the body. Thus, patients with facial pain are more sensitive to pain, experience more fear, and are more emotionally drained, which may result in a poorer response to treatment.

In general, KMT treatment sessions are important for the improvement of pain. One study [62] reported that one treatment session per week was ineffective in controlling neuropathic pain. Another study showed that the effect of pain reduction decreased when the frequency of treatment was changed from two sessions to one session per week [63]. Similarly, in the present study, patients who received treatment more than twice a week experienced a reduction in worst pain, which tended to be greater than that in patients who received treatment less than twice a week.

In our study, patients with PHN were mainly treated with acupuncture and BVA, and various other modalities such as HM, moxibustion, and cupping were also applied. It was expected that each KMT would be effective for PHN through the aforementioned mechanisms, and our results show a positive effect for reducing pain intensity and improving other conditions after integrative KMT on PHN. Future studies comparing the effectiveness of integrative KMT with each particular intervention are required.

There were unavoidable limitations to this study given that it was a retrospective chart review. First, only a small number of participants, 13, were included in the study. However, a study suggested that the number of participants required for the pilot study was at least 12 per group based on feasibility for block design, gain in precision about the mean and variation, and regulatory considerations [64]. Further studies with larger sample sizes are required to determine the appropriate effect size. Second, the long-term effects of KMT on PHN could not be analyzed because no records of long-term treatments of patients were retrieved. Third, since the integrative KMT was composed of multiple treatment modalities, it was not possible to differentiate which of the treatments had a greater effect on the improvement of PHN. However, it is important to use the synergetic effects of integrative KMT, which accurately reflects the clinical practice setting. Finally, no side effects were observed. Although KMT has few side effects, mild AEs such as bleeding, needle site pain, and subcutaneous hematoma may occur. Therefore, it is necessary to collect and analyze the data of AEs carefully. Based on this study, further well-designed studies are required to complement the aforementioned limitations, including a sufficient number of treatment sessions and long-term follow-ups.

## 5. Conclusions

In conclusion, the present study found that integrative KMT is effective not only in reducing pain, but also in improving anxiety, depression, sleep disturbance, fatigue, and QoL in patients with PHN. No AEs were observed. However, the results should be interpreted with caution because the study population was small and long-term treatment was not observed. Despite these limitations, this study shows the meaningful effectiveness of integrative KMT for PHN.

## Figures and Tables

**Figure 1 healthcare-12-00256-f001:**
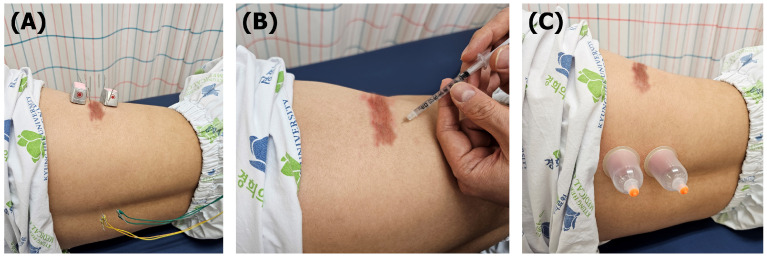
Integrative Korean medicine treatments for patient with postherpetic neuralgia. (**A**) Acupuncture and moxibustion treatment (43 °C) at local points and electroacupuncture at segmental points (100 Hz); (**B**) bee venom acupuncture treatment at local points; (**C**) cupping treatment at segmental points.

**Figure 2 healthcare-12-00256-f002:**
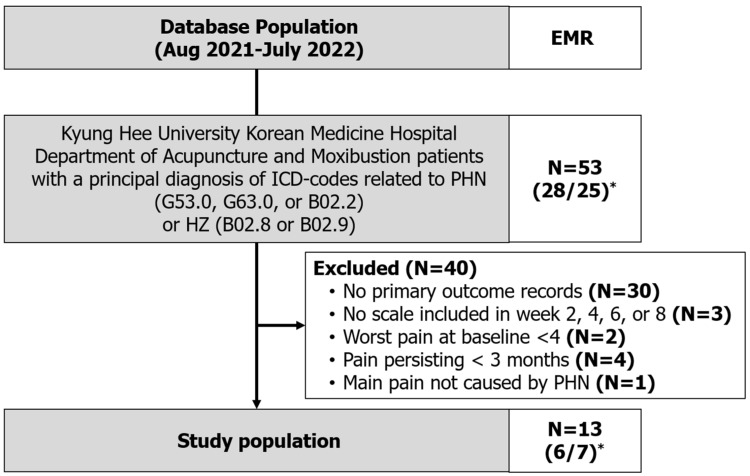
Flowchart of the included patients. * Number of patients (women/men).

**Figure 3 healthcare-12-00256-f003:**
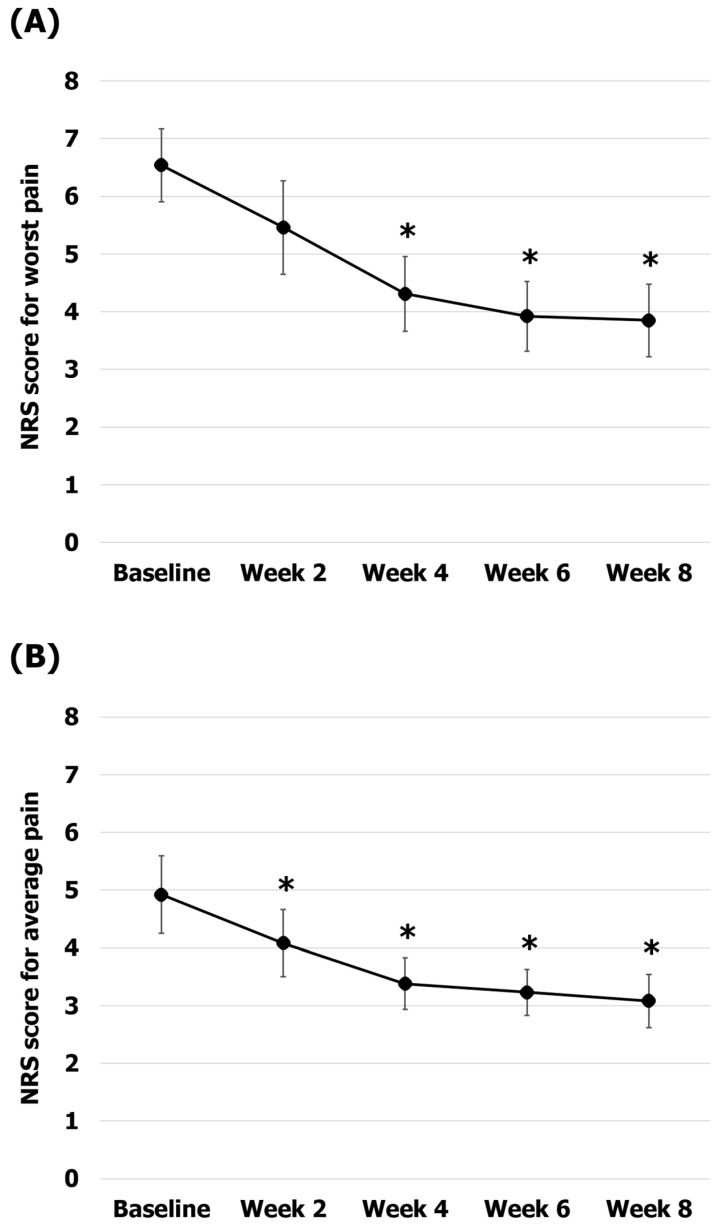
Changes in the pain scores during Korean medicine treatment. (**A**) The worst pain (NRS-W) scores; (**B**) the average pain (NRS-A) scores. The error bars represent the associated standard errors. * *p* < 0.05 compared to baseline. NRS-W, worst pain scores over 1 week on the numerical rating scale; NRS-A, average pain scores over 1 week on the numerical rating scale.

**Figure 4 healthcare-12-00256-f004:**
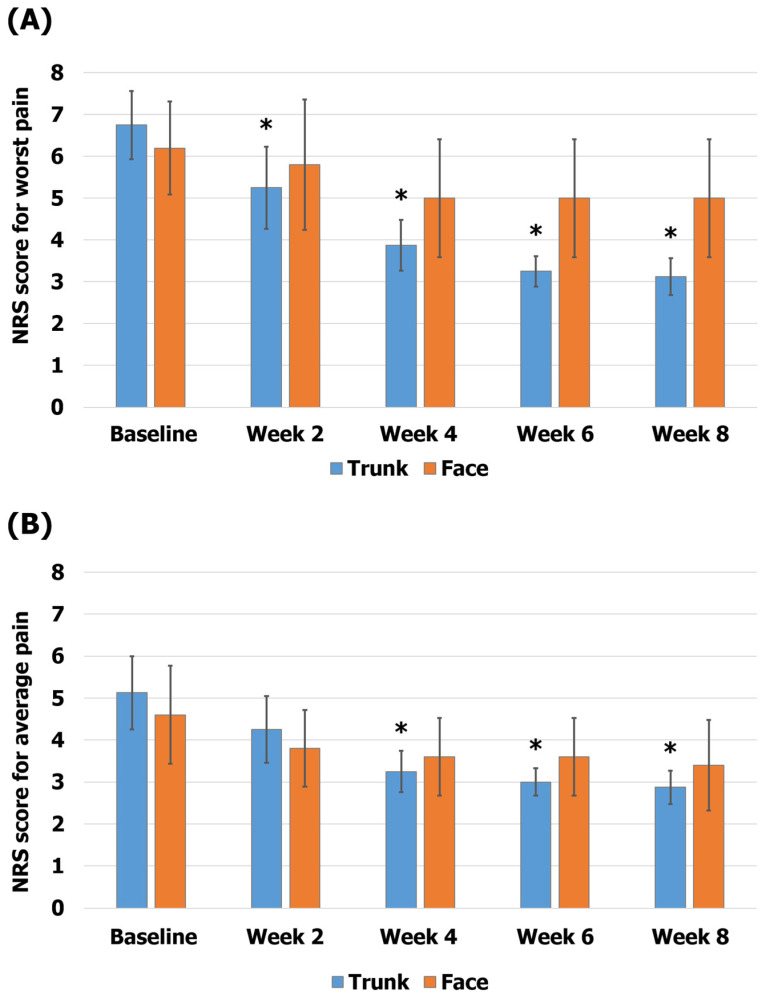
Comparison of pain according to the region of the rash. (**A**) The worst pain (NRS-W) scores; (**B**) the average pain (NRS-A) scores. The error bars represent the associated standard errors. * *p* < 0.05 compared to baseline. NRS-A, average pain scores over 1 week on the numerical rating scale; NRS-W, worst pain scores over 1 week on the numerical rating scale.

**Figure 5 healthcare-12-00256-f005:**
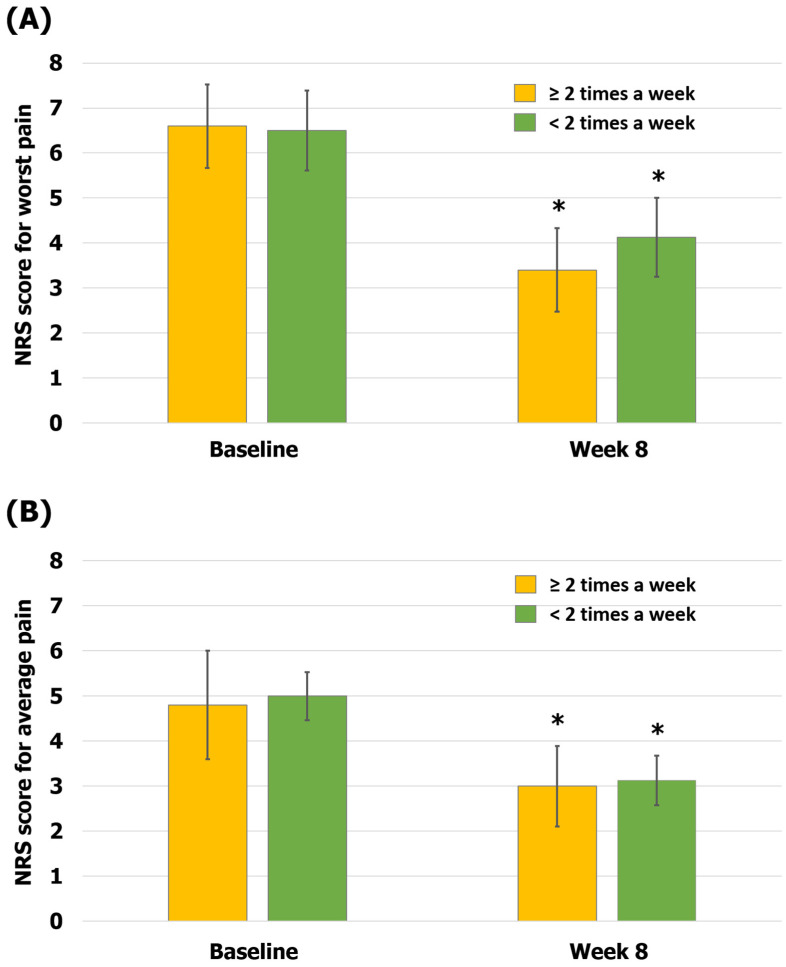
Comparison of pain according to the number of treatment sessions. (**A**) The worst pain (NRS-W) scores; (**B**) the average pain (NRS-A) scores. The error bars represent the associated standard errors. * *p* < 0.05 compared to baseline. NRS-A, average pain scores over 1 week on the numerical rating scale; NRS-W, worst pain scores over 1 week on the numerical rating scale.

**Table 1 healthcare-12-00256-t001:** Baseline characteristics of the study population.

Variable	Value
Age (mean ± SD), years	70.38 ± 10.68
Number (women/men)	13 (6/7)
Height (mean ± SD), cm	165.73 ± 8.72
Weight (mean ± SD), kg	63.86 ± 10.44
Hypertension (*n* [%])	7 (53.9)
Diabetes (*n* [%])	0 (0)
Dyslipidemia (*n* [%])	2 (15.4)
Cancer (*n* [%])	1 (7.7)
Medications (*n* [%])	13 (100)
Anticonvulsants	11 (84.6)
Antidepressants	3 (23.1)
Analgesics (non-opioids)	6 (46.2)
Opioids	2 (15.4)
Benzodiazepine	1 (7.7)
H/o intervention treatment * (*n* [%])	7 (53.8)
Pain duration (mean ± SD), months	27.08 ± 19.56
Rash region (*n* [%])	
Trunk	8 (62)
Face	5 (38)
Pain intensity NRS score over a week (mean ± SD)	(worst, average)
Trunk	(6.75 ± 2.31, 5.13 ± 2.47)
Face	(6.2 ± 2.5, 4.6 ± 2.61)
Number of treatment sessions in the first 4 weeks (mean ± SD)	9 ± 7.5
≥twice a week	15.6 ± 8.9
<twice a week	4.88 ± 0.64

H/o, history of; SD, standard deviation; n, number; NRS, numerical rating scale. Data are presented as mean ± SD unless otherwise indicated. * Intrathecal injection of methylprednisolone, peripheral nerve block, pulsed radiofrequency, and spinal cord stimulation were included.

**Table 2 healthcare-12-00256-t002:** Details of Korean medicine treatment prescriptions.

Korean Medicine Treatment Modality	Number (Percentage)
Acupuncture	13 (100)
Bee venom acupuncture	13 (100)
Herbal medicine	9 (69)
Cupping	10 (77)
Moxibustion	11 (85)

Data represent the number and percentage of patients who received treatment according to the Korean medicine treatment modality.

**Table 3 healthcare-12-00256-t003:** Changes in secondary outcomes during Korean medicine treatment.

Measurements	Baseline	Week 2	Week 4	Week 6	Week 8
SF-MPQ	33.85 ± 10.18	29.08 ± 7.6 * (14%)	27.54 ± 8.26 * (19%)	27.46 ± 8.61 * (19%)	26.08 ± 8.15 ** (23%)
SF-MPQ-S	26 ± 7.27	23.08 ± 5.65 * (11%)	21.85 ± 6.8 ** (16%)	21.77 ± 6.97 * (16%)	20.85 ± 7.15 ** (20%)
SF-MPQ-A	7.85 ± 3.24	6 ± 2.31 * (24%)	5.77 ± 1.96 * (26%)	5.7 ± 2.1 * (27%)	5.31 ± 1.55 * (32%)
HADS-A	8 ± 6.42	5 ± 4.58 (38%)	3.23 ± 2.71 ** (60%)	3 ± 2.55 ** (63%)	2.62 ± 2.22 ** (67%)
HADS-D	7.77 ± 5.36	5.92 ± 4.05 (24%)	6.15 ± 3.31 (21%)	6 ± 3.42 (23%)	5.54 ± 3.2 (29%)
DSIS	3.38 ± 2.72	2.77 ± 3.3 (18%)	2.23 ± 2.39 (34%)	2.31 ± 2.32 (32%)	2 ± 2.13 (41%)
FSS	3.7 ± 1.63	2.87 ± 1.17 (22%)	2.25 ± 1.3 ** (39%)	2.31 ± 1.29 ** (38%)	2.16 ± 1.38 ** (42%)
EQ-5D	5.31 ± 3.1	4.15 ± 2.44 (22%)	3.31 ± 1.75 * (38%)	3.31 ± 1.84 * (38%)	3.15 ± 1.72 * (41%)

SF-MPQ, Short-Form McGill Pain Questionnaire; SF-MPQ-S, Short-Form McGill Pain Questionnaire–Sensory dimension; SF-MPQ-A, Short-Form McGill Pain Questionnaire–Affective dimension; HADS-A, Hospital Anxiety and Depression Scale–Anxiety; HADS-D, Hospital Anxiety and Depression Scale–Depression; DSIS, Daily Sleep Interference Scale; FSS, Fatigue Severity Scale; EQ-5D, EuroQol-5D. Data are presented as the mean ± standard deviation (improvement %). * *p* < 0.05, ** *p* < 0.01 compared to baseline.

## Data Availability

The data presented in this study are available on request from the corresponding author.

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
