# Peer review of "Effect of Korean Medicine Treatment on Patients with Postherpetic Neuralgia: A Retrospective Chart Review"

_healthcare, 2024, doi:10.3390/healthcare12020256_

Round 1

Reviewer 1 Report

Comments and Suggestions for Authors

Comments to the authors

Thank you for inviting me to review the manuscript “Effect of a Korean medicine treatment on patients with postherpetic neuralgia: a retrospective chart review”. 

This is a retrospective study that looked at identifying effectiveness of a Korean medicine treatment on patients with PHN. Reportedly, the findings highlight potential significant reduction in average pain intensity and worst pain intensity on a cohort of 14 chart review patients. 

The exploration of Korean medicine treatment towards PHN is of high interest, considering that the current treatment options are limited and not always effective.

My main concern is that the sample size is very limited. On the top of that, the authors are comparing the effectiveness of multiple treatments (acupuncture, pharmacopuntcture, herbal medicine, cupping and moxibustion). Unless all of these treatments work through the same mechanism of action, otherwise how can the authors compare different treatments and merging them into one single category, in only 14 patients? I know that this has already been highlighted in the limitations of the study, but I wonder how valid the results are of the current study. I may suggest, as a way to get through it, to create a proof-of-concept for such small number patients treated with different treatment. Unless the authors really need to make a case that all of these treatments are similar and can be grouped together. 

Introduction: 

-       Authors should explain to the reader why in line 39 they are talking about shingles. 

-       Line 44: I would add pain-related symptomatology, as for example “fatigue” is a symptom and not a comorbidity 

-       The introduction may benefit from a brief explanation of what these treatments are and how they are supposed to work, especially if the authors are trying to make a case that these treatments can be categorized under one big treatment (KMT)

The authors have to provide details of these treatments. Are acupuncture, pharmacopuntcture, herbal medicine, cupping and moxibustion performed over the same period of time? By the same providers? With the same frequency? This is fundamental if we want to compare the effectiveness of a treatment (when the “treatment itself” is actually different among particiaptns!)

Methods: 

The description of the outcomes is very precise. However, does that mean that at every single point in time all these outcomes were assessed? Authors have to provide more details, such as if all these questionnaires were administered through in-person interview, or through email link, etc. Also, where these patients undergoing any medication to address PHN? 

Also, where the patients refractory to treatment? Were the traditional medications been tried already? 

Author Response

Introduction: 

-   Authors should explain to the reader why in line 39 they are talking about shingles. 

: We appreciate the reviewer's comments. We completely agree with the reviewer. In this paragraph, because shingles were used in the same sense as Herpes zoster, the existing word shingle was changed to Herpes zoster. (line 38)

-  Line 44: I would add pain-related symptomatology, as for example “fatigue” is a symptom and not a comorbidity 

: We appreciate the reviewer's comments. We completely agree with the reviewer. To reflect this, the word fatigue was deleted. (line 44)

-  The introduction may benefit from a brief explanation of what these treatments are and how they are supposed to work, especially if the authors are trying to make a case that these treatments can be categorized under one big treatment (KMT)

: We appreciate the reviewer's comments. We completely agree with the reviewer. Reflecting this, the treatment contents of KMT were described in detail, and an explanation of KMT was added in the Introduction session. (line 53-79)

The authors have to provide details of these treatments. Are acupuncture, pharmacopuntcture, herbal medicine, cupping and moxibustion performed over the same period of time? By the same providers? With the same frequency? This is fundamental if we want to compare the effectiveness of a treatment (when the “treatment itself” is actually different among particiaptns!)

: We appreciate the reviewer's comments. We completely agree with the reviewer. Patients with PHN were provided semi-standardized KMT twice a week. Taking this into account, additional descriptions about KMT were added to the Methods (line 102-144) and Result session (line 229-233)

Methods: 

- The description of the outcomes is very precise. However, does that mean that at every single point in time all these outcomes were assessed? Authors have to provide more details, such as if all these questionnaires were administered through in-person interviews, or through email link, etc.

: We appreciate the reviewer's comments. The Korean medical doctor assessed the outcomes by providing a pre-designed questionnaire, and the patient filled out all of the questionnaires in front of the doctor. If patients had questions about the questionnaire, they asked a Korean medical doctor and filled it out. We added additional descriptions to the Method session (line 148-151).

- Also, where these patients undergoing any medication to address PHN? Also, where the patients refractory to treatment? Were the traditional medications been tried already? 

: We appreciate the reviewer's comments. We added details of conventional medications to Table 1. And all patients received pharmacotherapy more than 3 months before receiving KMT for treating symptoms of PHN. We added additional descriptions to the Results (line 229- 230, Table 1).

Reviewer 2 Report

Comments and Suggestions for Authors

Manuscript shows a retrospective study using Korean Medicine Treatment in postherpetic neuralgia. Please, find here some considerations.

Line 123: ¿adverse effects or adverse events? Please use the same terminology

Line 125: use the AE abbreviation

Methodology: which are the treatment followed? Who implement this treatment? How often or how long the session takes?

Results: all treatments implemented were applied at the same frequency or intensity?  Could it be a bias?

Discussion:

Line 306-319: should be placed in the introduction section.

Is there any difference between the treatments that patients received?

Author Response

Comments to the authors

Manuscript shows a retrospective study using Korean Medicine Treatment in postherpetic neuralgia. Please, find here some considerations.

-Line 123: ¿adverse effects or adverse events? Please use the same terminology

Line 125: use the AE abbreviation

: We appreciate the reviewer's comments. We completely agree with the reviewer. To reflect this content, all terms were unified into adverse events and described using abbreviations.

-Methodology: which are the treatment followed? Who implement this treatment? How often or how long the session takes?

: We appreciate the reviewer's comments. Patients with PHN were provided semi-standardized KMT protocol twice a week and one Korean medical doctor (S Lee) treated all patients in this study. Taking this into account, additional descriptions about KMT were added to the Methods (line 102-144) and Result session (line 229-233).

Results

-all treatments implemented were applied at the same frequency or intensity?  Could it be a bias?

: We appreciate the reviewer's comments. Patients with PHN were provided semi-standardized KMT protocol twice a week. Although acupoints may vary slightly from patient to patient, the general principles (number of sessions, frequency of electroacupuncture, etc.) were provided to patients according to the defined KMT protocol. In some cases, patients who did not necessarily need herbal medicine or cupping treatment did not receive this treatment. Details of the semi-standardized KMT protocol and actual treatment received are detailed in the Methods (line 102-144) and Result session (line 229-233).

Discussion

-Line 306-319: should be placed in the introduction section.

: We appreciate the reviewer's comments. We completely agree with the reviewer. We reflected on this and moved the content to the introduction section. (line 53-79)

-Is there any difference between the treatments that patients received?

: We appreciate the reviewer's comments. Patients with PHN were provided semi-standardized KMT protocol twice a week. Although acupoints may vary slightly from patient to patient, the general principles (number of sessions, frequency of electroacupuncture, etc.) were provided to patients according to the defined KMT protocol. In some cases, patients who did not necessarily need herbal medicine or cupping treatment did not receive this treatment. Details of the actual treatment received are detailed in the Result session (line 229-233, Table 2).

Round 2

Reviewer 1 Report

Comments and Suggestions for Authors

Comments to the authors

The authors have provided an updated version that attempted to reflect previous requests of the reviewer. The authors have done a very good job and the study increased in quality. Still, there are some points that need to be addressed to achieve the full potentiality of the paper. Here’s my new comments to the present version of the manuscript. 

Introduction:

Line 58-59: if acupuncture, pharmacopuntuce and HM are effective in improving pain, they are likely to increase the secretion of endogenous opioid peptides, serotonin, norepinephrine, rather than decreasing their release. Also, it would be very interesting if the authors could elaborate more on the mechanism of actions through which acupuncture, and other techniques work. The increase in endogenous opioid peptides, serotonin and NE is not the only mechanism. For example, https://pubmed.ncbi.nlm.nih.gov/32420752/#:~:text=Acupuncture%20reduces%20pain%20by%20activating,felt%20by%20clinicians%20and%20patients.

https://pubmed.ncbi.nlm.nih.gov/18582529/

And others. Also, the DNIC should be mentioned as possible mechanism of action. 

Line 79-80: the aim should clearly state what “effect” the authors are investigating: effect on pain outcomes? Pain intensity? Frequency? Psychological distress? “Effect” is very very vague and unspecific. 

The authors should also anticipate their working hypothesis after the aims. 

Methods: this section improved dramatically since the original manuscript. If the authors could add some pictures to display the different treatments, that would increase the readability and impact of the article. 

Lines 132-136: is it possible to clarify the range of strength of the Herbal medicine? 

Secondary and Primary outcomes: the outcomes should not be the scale / questionnaires used to measure the real outcome. The outcomes should be for example, average pain intensity (primary), anxiety symptomatology, etc. The authors should clarify the outcomes, and then clarify how the outcomes was assessed (questionnaire or scale). 

Line 201: here again the term “shingles”. 

It may be a personal preference, but the 2.5 Data collection should be put before the outcomes. 

Statistical analysis: this only clarify what tests were used if the data were not normally distributed. What if they were normally distributed? And this section should clearly state which groups were compared and on what variables, and through which test. The missing data sentence should be put before commenting on the data distribution. 

Lines 384: here the authors briefly mention about the required sample size. I suggest that the authors devote a paragrpha in the methods to talk about this minimum required sample size (how it was achieved by commenting the study they quote). This would increase validity of the results. 

Author Response

Introduction:

  • Line 58-59: if acupuncture, pharmacopuntuce and HM are effective in improving pain, they are likely to increase the secretion of endogenous opioid peptides, serotonin, norepinephrine, rather than decreasing their release. Also, it would be very interesting if the authors could elaborate more on the mechanism of actions through which acupuncture, and other techniques work. The increase in endogenous opioid peptides, serotonin and NE is not the only mechanism. For example, https://pubmed.ncbi.nlm.nih.gov/32420752/#:~:text=Acupuncture%20reduces%20pain%20by%20activating,felt%20by%20clinicians%20and%20patients. https://pubmed.ncbi.nlm.nih.gov/18582529/

And others. Also, the DNIC should be mentioned as possible mechanism of action.

: We appreciate the reviewer's comments. We completely agree with the reviewer. Referring to the advice you provided, we have added detailed information about the related treatment mechanism in the Introduction section. (line 56-63)

  • Line 79-80: the aim should clearly state what “effect” the authors are investigating: effect on pain outcomes? Pain intensity? Frequency? Psychological distress? “Effect” is very very vague and unspecific. The authors should also anticipate their working hypothesis after the aims.

: We appreciate the reviewer's comments. We completely agree with the reviewer. Based on your advice, we have presented more specific goals. (line 82-85)

Methods: this section improved dramatically since the original manuscript.

  • If the authors could add some pictures to display the different treatments, that would increase the readability and impact of the article.

: We appreciate the reviewer's comments. We completely agree with the reviewer. Based on the advice you provided, we added a figure of Korean medicine treatment for patients with postherpetic neuralgia. (Figure 1)

  • Lines 132-136: is it possible to clarify the range of strength of the Herbal medicine?

: Thank you for the reviewer's comments. We completely agree with the reviewer. We have added information related to herbal medicine as requested. (Line 143-145)

  • Secondary and Primary outcomes: the outcomes should not be the scale/questionnaires used to measure the real outcome. The outcomes should be, for example, average pain intensity (primary), anxiety symptomatology, etc. The authors should clarify the outcomes, and then clarify how the outcomes were assessed (questionnaire or scale).

: We appreciate the reviewer's comments. We completely agree with the reviewer. Based on your review, we have clearly revised the primary and secondary outcomes. We also described in detail how to evaluate these outcomes. (line 167-210)

  • Line 201: here again the term “shingles”.

: We appreciate the reviewer's comments. We completely agree with the reviewer. To reflect this, the meaning of 'shingle' in the text was changed to 'rash' as it could cause confusion. (line 250, 382,383)

  • It may be a personal preference, but the 2.5 Data collection should be put before the outcomes.

: We appreciate the reviewer's comments. We completely agree with the reviewer. As you commented, the ‘Data collection’ part was placed before the ‘Outcomes’ part.

  • Statistical analysis: this only clarify what tests were used if the data were not normally distributed. What if they were normally distributed? And this section should clearly state which groups were compared and on what variables, and through which test. The missing data sentence should be put before commenting on the data distribution.

: Thank you for the reviewer's comments. We completely agree with the reviewer. Statistical information has been added to reflect the review results. (Line 232-236)

  • Lines 384: here the authors briefly mention about the required sample size. I suggest that the authors devote a paragrpha in the methods to talk about this minimum required sample size (how it was achieved by commenting the study they quote). This would increase validity of the results.

: Thank you for the reviewer's comments. We completely agree with the reviewer. Reflecting on your review, we have additionally described the minimum required sample size. (Line 406-409)

Reviewer 2 Report

Comments and Suggestions for Authors

Thank you for following my suggestions.

Author Response

Thanks to your great review, the quality of our manuscript has improved significantly. Thank you so much.